# Improving Soy Sauce Aroma Using High Hydrostatic Pressure and the Preliminary Mechanism

**DOI:** 10.3390/foods11152190

**Published:** 2022-07-23

**Authors:** Yaqiong Zhang, Zhi-Hong Zhang, Ronghai He, Riyi Xu, Lei Zhang, Xianli Gao

**Affiliations:** 1School of Food and Biological Engineering, Jiangsu University, Zhenjiang 212013, China; 2222018115@stmail.ujs.edu.cn (Y.Z.); zhihong1942@foxmail.com (Z.-H.Z.); heronghai@163.com (R.H.); zhangleifd@ujs.edu.cn (L.Z.); 2Institute of Biological and Medical Engineering, Guangdong Academy of Sciences, Guangzhou 510316, China; scutriyal@163.com

**Keywords:** soy sauce, high hydrostatic pressure, aroma compound, *Aspergillus oryzae*, soybean

## Abstract

Using high hydrostatic pressure (HHP) to treat liquid foods can improve their aroma; however, no information about the effects of HHP on soy sauce aroma has yet been reported. The effects of HHP on the aroma of soy sauce fermented for 30 d were investigated using quantitative descriptive analysis (QDA), SPME–GC–olfactometry/MS, hierarchical cluster analysis (HCA) and principal component analysis (PCA). Results showed that the pressure used during HHP treatment had a greater influence on soy sauce aroma than the duration of HHP. Compared to the control, soy sauce that was treated with HHP at 400 MPa for 30 min (HHP400–30) obtained the highest sensory score (33% higher) by increasing its sour (7%), malty (9%), floral (27%) and caramel-like (47%) aromas, while decreasing its alcoholic (6%), fruity (6%) and smoky (12%) aromas; moreover, the aroma of HHP400–30 soy sauce was comparable with that of soy sauce fermented for 180 d. Further investigation demonstrated that HHP (400 MPa/30 min) enhanced the OAVs of compounds with sour (19%), malty (37%), floral (37%), caramel-like (49%) and other aromas (118%), and lowered the OAVs of compounds with alcoholic (5%), fruity (12%) and smoky (17%) aromas. These results were consistent with the results of the QDA. HHP treatment positively regulated the Maillard, oxidation and hydrolysis reactions in raw soy sauce, which resulted in the improvement and accelerated formation of raw soy sauce aroma. HHP was capable of simultaneously improving raw soy sauce aroma while accelerating its aroma formation, and this could treatment become a new alternative process involved in the production of high-quality soy sauce.

## 1. Introduction

Soy sauce is a popular fermented liquid condiment with an attractive aroma and umami taste that is gaining popularity around the world [1]. Soy sauce is manufactured using soybeans/defatted soybean flakes, roasted wheat, raw wheat flour, salt and water as materials, with *Aspergillus oryz**ae*, yeast and lactic acid bacteria used as starters. Its manufacturing process includes two necessary steps: koji fermentation and the following moromi fermentation. Koji fermentation involves the inoculation of *A. oryzae* into the mixed steamed soybeans/defatted soybean flakes and raw wheat flour; subsequently, the inoculated mixture is cultured at approximately 30 °C for 44–48 h in order to make koji. The mature koji is blended with about 1–3-fold saline (18–22%), in order to make moromi. In China, the perfect aroma is achieved by moromi fermentation, which usually lasts for six months [2,3].

Free amino acids, small peptides, organic acids, sugars and NaCl determine the taste of soy sauce, which is an important attribute in consumers’ acceptance and preference for soy sauce. Studies have found that the aroma and taste of soy sauce are formed asynchronously [4]. The taste compounds in soy sauce reache their highest levels after fermenting for approximately 30 days. However, after that time the aroma quality is still considered to be far below the standard of a ‘perfect’ aroma of soy sauce [1,5]. The delayed aroma formation will inevitably decrease the production efficiency and profit of soy sauce manufacturers; thus, it would be of great importance to expedite the aroma maturation of soy sauce.

The traditional fermentation industry typically faces challenges that involve large production areas and long fermentation times. Thus far, some new technologies, i.e., sonication and HHP, have been investigated in order to accelerate the maturation of fermentation products [6,7,8]. Results demonstrated that ultrasound could be used to accelerate the aroma formation of wine [9], vinegar [10,11] and soy sauce [3], while shortening the fermentation time. Unfortunately, the acidic, alkaline or high-salt environment of foods could seriously corrode the ultrasonic equipment when they are in direct contact with each other.

HHP equipment is capable of treating packaged foods by using pressure-tolerant bags, consequently avoiding direct contact between the food and equipment [12]. Thus, HHP can become a potential alternative method that could be used to accelerate soy sauce aroma formation. HHP treatment, as a non-thermal processing technology, can instantly and uniformly transfer pressures of 100–1000 MPa throughout the food [12,13,14]. HHP is considered to be a green technology because it utilizes water as a medium to transfer pressure, and does not require energy to maintain the high pressure once it is reached. Currently, HHP equipment has been successfully used in the processing of fruit juice, beverages, vegetables, seafood, dairy products, rice products, meat products as well as for other products [15,16,17]. A study reported that HHP treatment (400 MPa) accelerated the aging process of Cayetana white wine, which only took 10 min to achieve physicochemical and sensory characteristics similar to wine treated with traditional oak barrel fermentation for 45 days [16]. In addition, researchers found that HHP treatment increased the level of polyphenols in white wine. Tian et al. found that the volatile compounds and sensory quality of Hongqu rice wine that had been treated for 30 min at 550 MPa and then stored for 6 months were similar to those of untreated rice wine kept in pottery for 18 months [18].

In accordance with Le Chatelier’s principle, a decrease in the volume of the reaction system caused by HHP will lead to a change in the equilibrium of a chemical reaction [12,19]. Therefore, HHP may alter the balance of chemical reactions in soy sauce, and thereby accelerate the formation of aromatic compounds, rapidly change the sensory characteristics of soy sauce, and thus accelerate the formation of soy sauce aroma compounds. The above process can also inactivate unwanted microorganisms in soy sauce [15,17]. Therefore, HHP treatment may be a potential method for accelerating the maturation of soy sauce aroma and bring additional benefits to soy sauce manufacturers. However, the utilization of HHP in soy sauce industry has not been investigated in-depth until now.

The objectives of this paper are to explore the influences of HHP treatment on the aroma compounds and aroma of soy sauce, and to elucidate the preliminary mechanism of the rapid aroma maturation of soy sauce caused by HHP treatment.

## 2. Materials and Methods

### 2.1. Materials and Chemicals

Soybeans, raw wheat flour and edible salt (NaCl) were purchased from a local supermarket (Zhenjiang, China, 2021). 3-Methylbutanal, ethanol, 3-methyl-1-butanol, ethyl propanoate, ethyl acetate, ethyl isovalerate, 2-phenylethanol, benzeneacetaldehyde, 3-(methylthio) propyl acetate, ethyl 3-(methylthio) propionate, methyl benzoate, (E)-β-damascenone, 2-phenylethyl acetate, 2-furanmethanol, 3-methylthio-1-propanol, 3-methylthio propanal, 2,6-dimethyl pyrazine, trimethyl pyrazine, 2-acetyl pyrroline, 4-hydroxy-2,5-dimethyl-3(2H)-furanone (HDMF) and 4-hydroxy-2(or 5)-ethyl-5(or 2)-methyl-3(2H)-furanone (HEMF) were all ordered from Sigma-Aldrich (Shanghai, China, 2021). 3-Furaldehyde, 4-ethylphenol, 4-ethylguaiacol, 4-vinylguaiacol, 3-methylbutanoic acid, acetic acid, 2-methylbutanoic acid, dimethyl trisulfide, 1-octen-3-ol and C6-C33 n-alkanes were provided by Aladdin Holdings Group (Shanghai, China, 2021). Other analytically pure reagents were purchased from Sinopharm Chemical Reagent Co., Ltd. (Shanghai, China, 2021). *A. oryzae* 3.042 were obtained from the Guangdong Institute of Microbiology (Guangzhou, China, 2021).

### 2.2. Soy Sauce Preparation and HHP Treatment

Soy sauce was prepared according to the approach described by Gao et al., with some modifications [3]. Soybeans and raw wheat four were used to prepare koji using *A. oryzae* as the starter; then, the koji and 2.2-fold saline (22%, *w*/*v*) were mixed to prepare moromi, with the latter being sealed in a 5-L stainless steel fermenter and fermented at room temperature. Since the taste was mature on approximately the 30th day, raw soy sauces were sampled on the 30th day and then stored at −20 °C in a freezer for analyses.

The raw soy sauce was thawed and filtered through qualitative filter paper. Each aliquot of filtrate (50 mL) was sealed in a sterilized polyethylene bag. After sealing, the samples were placed in HHP equipment (S-HPP-5L, LDF, Taiyuan, China) and treated with pressures of 200, 400 and 600 MPa using water as a medium; each pressure was maintained for 10, 30 and 60 min, respectively. All of the treatments were conducted at ambient temperature (approximately 25 °C). The temperature during the pressurization process did not exceed 35 °C. All samples as well as the control were stored in a 4 °C refrigerator until analyses were conducted.

Soy sauce samples that were fermented for 30 d without HHP treatment were designated as the control. HHP200–10 represented soy sauce that was treated with HHP for 10 min at 200 MPa. The remaining samples were correspondingly designated as HHP200–30, HHP200–60, HHP400–10, HHP400–30, HHP400–60, HHP600–10, HHP600–30 and HHP600–60.

### 2.3. Sensory Evaluation

A quantitative descriptive analysis (QDA) was performed according to the method described in a previous study [1]. In total, nine sensory evaluators aged 23–43 years including five males and four females from Jiangsu University with normal olfactory sensation were recruited as sensory panelists. The evaluation was performed according to the previous description by Gao et al., (2010) [1].

### 2.4. Volatile Compounds Extraction Using Solid-Phase Micro-Extraction (SPME)

In order to verify the effects of HHP treatment on aroma compounds in soy sauce, samples (10 mL) and controls (10 mL) were saturated using NaCl, and then sealed in 50-milliliter headspace bottles. SPME fiber (Supelco, Bellefonte, PA, USA) was coated with 75-micrometer carboxen-polydimethylsiloxane coating and pretreated at 275 °C and 250 °C for 1 h and 30 min, respectively, in a GC sampler; this was done to remove the residues. Prior to extraction, the headspace bottle was preheated for 10 min at 50 °C while being stirred with a magnetic stirring bar at a speed of 200 rpm. Then, SPME fiber was used to adsorb the volatile compounds for 40 min at 50 °C.

### 2.5. Analyses of GC-Olfactometry (GC-O) and Flavor Dilution (FD) Factor

The aroma-active compounds in the samples and control were screened using GC-O identification with an Agilent 6890 gas chromatograph equipped with an Agilent 5973 N mass selection detector (Wilmington, DE, USA) and a sniffing port (ODP-2; Gerstel, Inc., Linthicum, MD, USA), based on the previous description of Gao et al., (2020) [3].

The flavor dilution factor (FD) represents the highest split ratio (most dilution) of GC injection at which the odorant could be noticed (even if not identified) by at least two out of three panelists [3]. In this study, the split ratios of GC injection were 1:1, 2:1, 4:1, 8:1, 16:1, 32:1, 64:1, 128:1, 256:1 and 512:1.

### 2.6. Characterization and Quantification of Volatile Compounds

The above-mentioned GC-MS system as well as a DB-Wax column (30 m × 0.25 mm i.d × 0.25 μm film thickness; J&W Science, Folsom, CA, USA) was utilized to isolate the volatiles. Helium was used to elute the volatiles under a constant flow rate of 1.0 mL/min and a split ratio of 15:1. The oven was programmed with a heating speed of 5 °C/min, from 40 °C (held for 2 min) to 120 °C (held for 2 min), then ramped up to 230 °C with a speed of 7 °C/min, before being finally maintained at 230 °C for 4 min. The SPME fiber was injected into the GC sampler and the extracted volatiles were desorbed at 250 °C for 5 min. Subsequently, the SPME fiber was kept in the GC injector for another 5 min in order to completely remove the residues on the fiber. The conditions of mass spectrometry used were as follows: 250 °C for the temperature of the ion source, 70 eV for the electron energy, and 30–450 *m*/*z* for the mass scanning range.

Volatile compounds were identified according to the Kovats Retention Inde (RI) and through matching mass spectra from the NIST05 library. In addition, C6-C33 n-alkanes were used as standards to determine RIs under the same GC conditions. The external standard method was used to quantify all aroma-active compounds. The odor activity value (OAV) of each aroma-active compound was calculated using the corresponding threshold value to divide its concentration [20].

### 2.7. Statistical Analysis

In this study, all tests except sensory tests were repeated thrice, and all data were presented as mean ± standard deviation (SD). SPSS 15.0 software (SPSS Inc., Chicago, IL, USA) was utilized for one-way ANOVA to determine that the differences were statistically significant within a 95% confidence interval, and the significance level was set to *p* < 0.05. Principal component analysis (PCA) and hierarchical cluster analysis (HCA) were conducted using Origin 8.6 (OriginLab Corporation, Northampton, MA, USA) and SPSS 15.0 using OAVs as variables.

## 3. Results and Discussion

### 3.1. Sensory Evaluation

Sensory analysis is the most direct method used to evaluate the quality of soy sauce. We conducted a QDA of soy sauce to explore the effects of HHP treatment on its sensory characteristics. As shown in Figure 1, overall, the aroma scores of sour, caramel-like, malty and floral in soy sauces treated using HHP increased by 1.05–7.37%, 13.33–48.89%, 0.43–8.70% and 0.91–27.27% respectively, while the alcoholic, fruity and smoky aromas decreased by 0.56–11.11%, 1.29–12.90% and 0.77–32.31%, respectively, compared to the control. In terms of overall sensory evaluation scores, the overall scores of the 200 MPa treatment group and the 400 MPa treatment group were 9.37–14.38% and 21.25–32.81% higher than that of the control, respectively. In the 600 MPa treatment group, only HHP600–10 increased by 10.94% compared with the control, while the scores of HHP600–30 and HHP600–60 decreased by 4.69% and 9.38%, respectively. Among the sensory scores of sour, caramel-like, malty and floral, those of HHP400–30 increased by 7.37%, 46.67%, 8.70% and 27.27%, respectively, while the alcoholic, fruity and smoky aromas decreased by 5.56%, 6.45% and 11.54%, respectively, compared to the control. HHP400–30 obtained the highest overall sensory evaluation score of 4.25, which was higher than that of soy sauce fermented for 180 d [3]. These results indicated that proper HHP treatment could simultaneously accelerate maturation of soy sauce aroma and improve its aroma quality.

Among all the aromas, caramel-like aroma was characterized with the best improvement effect, which may be a result of the HHP treatment accelerating the generation of products of the Maillard reaction in soy sauce [19,21,22]. Although HHP400–30 was less alcoholic, fruity and smoky, its soft, coordinated aroma was more acceptable. Its aroma was defined by sensory evaluators as sweeter, softer, more harmonious and more complex, since it contained a greater variety of aromatic substances, as demonstrated in Appendix A. This sauce has the potential to become a new kind of aromatic soy sauce. A study of wine treated with HHP (400 MPa, 5 min and 400 MPa, 30 min) found that the treated wine was defined by tasters as “more complex wines” with “toasted” aroma descriptors [16]. Another study found that the aromas of rice wines treated with HHP (550 MPa, 30 min) which were stored for 6 months were very similar to that of the untreated rice wine stored for 18 months [18]. These studies showed that appropriate HHP treatment could accelerate wine aging and improve wine sensory quality, which is consistent with the conclusion of this study.

### 3.2. Characterization of Volatile Compounds

As demonstrated in Table 1, 133 volatile compounds, including 27 alcohols, 13 acids, 14 ketones, 8 aldehydes, 10 furan(one)s, 32 esters, 9 pyrazines, 7 phenols, 7 sulfur-containing compounds in addition to 6 other compounds were characterized in the HHP-treated samples and their controls. Compared with our previous studies, more alcohols, ketones, esters, phenols and pyrazines were identified in this study. This may be attributed to differences in the samples used in previous studies, but the kinds of main aroma compounds in soy sauce are similar to those previously reported by our research group [1,3].

Notably, two sulfur-containing compounds, 3-(methylthio) propyl acetate and ethyl 3-(methylthio) propionate, were first identified in soy sauce; their aromas were described as fruity, sulfur and garlic, and meaty, onion and fruity, respectively [23,24]. Since these two substances were identified only in HHP-treated samples, it was inferred that HHP treatment promoted the oxidation of 3-(methylthio) propanol and 3-(methothio) propanal to produce 3-(methothio) propanic acid, which promoted the esterification reaction of 3-methothiopropanic acid with ethanol to form ethyl 3-(methylthio) propionate. Meanwhile, part of 3-(methylthio) propanol reacted with acetic acid to form 3-(methylthio) propyl acetate. Due to its OAV (greater than 1), ethyl 3-(methylthio) propionate was defined as a main aroma-active compound, and it has also been reported to be an aroma-active compound in pineapple by Xiao et al., (2021) [23]. However, 3-(methylthio) propyl acetate was identified in wine with ignorable influence on aroma by Moreira et al., (2010) [24]. Our results also showed that there was only a little 3-(methylthio) propyl acetate in soy sauce, hence it was not a main aroma-active compound.

Acids contribute to the sour aroma of soy sauce, while other compounds such as benzoic acid provide floral and fruity aromas. A total of 13 acids were identified in this work; except for 2-methylbutanoic acid, the other acids increased to different degrees under HHP treatment (Appendix A). However, due to the high odor threshold of the main aroma compounds, for example acetic acid (10,000), 2-methylpropanoic acid (3500) and 3-methylbutanoic acid (540), the increase in acids did not lead to a significant improvement in the sensory evaluation of acid aroma, which only increased by 7.37% in HHP400–30 (Figure 1). Xia and Li treated whole grain brown rice with HHP for 10 min at 150 MPa, 350 MPa and 450 MPa, and also found that the total acid content in wholegrain brown rice was enhanced, and that the total acid content increased with an increase in pressure [25].

More alcohols were identified in this study; overall, the contents of high-carbon alcohols above C5 in soy sauce after HHP400–30 treatment increased significantly; apart from 4-methyl-2-pentanol, 1-nonanol, phenylethanol and α-ethyl phenylethanol, the contents of 17 high-carbon alcohols all increased (Appendix A). An important reason for this could be that HHP treatment makes the chemical reactions move in the direction of volume reduction, consequently providing the conditions for aldol condensation, which leads to the condensation of low-carbon alcohol to form more high-carbon alcohols [12,13]. In addition, the increases in various high-carbon alcohols, such as hexanol (green, floral), 2-nonanol (cucumber), 3-octanol (nut, mushroom), 1-octene-3-ol (mushroom), benzyl alcohol (sweet, flower) and phenylethanol (flower, honey), bring richer flavor to soy sauce. However, the total alcohol content decreased by 7.49% (Appendix A), among which the contents of 10 alcohols that have fewer than or equal to five carbons decreased, except for 2-butanol, 3-methyl-1-butanol and prenol. Especially, the content of ethanol decreased by 7.40% (Appendix A), which was the main contributor to the decrease in total alcohol content. 2-Methyl-1-butanol, ethanol and 3-methyl-1-butanol were the main contributors to alcoholic aroma, and the reduction in the content of ethanol and 2-methyl-1-butanol correspondingly led to the reduction in alcoholic aroma in sensory evaluations. The reduction in alcohols due to HHP treatment was also observed in the identification of aroma compounds in various fruit products [26,27,28]. The decrease in the presence these alcohols may be due to their higher volatility or to the accelerated oxidation reaction under HHP treatment [13], resulting in more acids, which also well explains the increase in acids in soy sauce observed after HHP treatment.

A total of 8 aldehydes and 14 ketones were detected in this study; only acetaldehyde, benzaldehyde, acetoin, 2, 3-pentanedione and 1-phenyl-1-propanone contents decreased or were not detected in HHP400–30. For aldehydes, three substances (2-methylbutanal, 2-methylpropanal, 3-methylbutanal) are the chief ingredients in soy sauce that produce a malty aroma [3,5,29]. The total contents of aldehydes and ketones in the soy sauce treated with HHP (400 MPa, 30 min) were 1.30 times and 2.35 times (Appendix A) of that of the control, respectively. Increased aldehydes and ketones were also found in germinated brown rice [30], green asparagus juice [26] and mulberry juice [31], as well as in oysters [19] and wholegrain brown rice [25] after HHP treatment. Aldehydes and ketones can be produced by lipid oxidation or through amino acid degradation induced by the Strecker reaction [13]. Soy sauce, a protein-containing matrix, may be affected by HHP treatment because it is rich in the free amino acids required for the formation of aldehydes and ketones [2].

In this study, a total of 32 esters were detected, of which only 7 were improved; however, the total ester content decreased by 15.06% (Appendix A). Except for ethyl 2-ethylhexanoate, the contents of 25 ethyl esters decreased by differing degrees. The decrease in ester compounds may be due to the hydrolysis reaction of esters, which was enhanced by the HHP treatment [13]. Santos et al. found that HHP treatment reduced the content of aliphatic ethyl esters in white wine [21]. Similarly, the same tendency was found in red plum puree [27], germinated brown rice [30] and kiwi fruit pulp beverage [32] treated under HHP. Esters are the main contributors of fruit aroma in soy sauce [3]; consequently, a decrease in their content also resulted in a decrease in fruity aroma scores in the sensory evaluations. However, due to the high aroma threshold of the main compound ethyl acetate (5000), fruity aroma only decreased by 6.45% in HHP400–30 (Figure 1).

Phenol compounds have a unique phenol flavor and smoky aroma, and are important compounds in producing the unique flavor of soy sauce [3]. It is worth mentioning that studies have shown that more complex phenolic compounds (such as 4-methylguaiacol 4-ethylguaiacol and 4-vinylguaiacol) have more desirable aroma characteristics than simple phenolic compounds [13]. The HHP treatment reduced the contents of three kinds of phenols (4-ethylguaiacol, 4-ethylphenol and 4-vinylguaiacol) that produce a smoky aroma; the treatment resulted in a 36.7% reduction in total phenol content (Appendix A). The concentration of phenols compounds in the matrix was a result of polyphenol oxidase (PPO) activity. If the activity of this enzyme is increased, the amounts of phenolic compounds will be reduced; the opposite will occur if it is inhibited. The decrease in phenolic compound content in this study may be due to PPO activity that was enhanced by the HHP treatment [33]. Since weak intramolecular bond interactions (chiefly electrostatic and hydrophobic interactions) will be damaged under HHP, this can lead to subsequent changes in the secondary, tertiary and quaternary structures of some enzymes, hence altering the concentrations of the corresponding compounds [13]. Similarly, the HHP treatment of green asparagus juice [26] and red wine rich in phenolic compounds [34] also resulted in reductions in phenolic compounds.

In sensory analysis experiments, we thought that the caramel aroma enhancement was due to the HHP processing that accelerated the Maillard reaction [19,21]. The aromatic compounds formed by Maillard reactions are categorized into three types: nitrogen containing compounds (pyrrolines and pyrazines), oxygen containing compounds (ketones, aldehydes, furanones) and sulfur containing compounds (thiazolines and thiazoles, dithiazoles, furansthiols and sulfides) [22]. Our team detected eight kinds of aldehydes, fourteen kinds of ketones, seven kinds of furans, three kinds of furanones, nine kinds of pyrazines and seven kinds of sulfur-containing compounds in this study. Except for two aldehydes (acetaldehyde and benzaldehyde), three kinds of ketones (acetoin, 2,3-pentanedione and 1-phenyl-1-propanone) and one pyrazine (2-ethyl-3,5-dimethylpyrazine), the contents of the other forty-two compounds were increased (Appendix A). Santos et al. found higher contents of aldehydes, furans, acetals and ketones in pressurized wines than those in unpressurized wines after 9 months of storage [21]. The changes in these volatile components suggested that the Maillard reaction could be promoted by HHP treatment.

### 3.3. Analyses of GC-O and FD Factor

As demonstrated in Table 1, 43 aroma-active compounds were detected by GC-O from 133 volatile compounds in the HHP-treated samples and control. The aroma-active compounds in the control and samples included seven acids, six esters, five alcohols, five aldehydes, four pyrazines, four furan(one)s, five sulfur-containing compounds, three phenols, three ketones and one pyrroline. In the previous study of our team, the aroma-active compounds in both samples treated with ultrasound (S180) and samples untreated with ultrasound (C180) were almost the same, except that there was no 2-phenylethyl acetate in C180 [3]. However, the types and distributions of aroma-active compounds in the HHP group and in the control without HHP treatment were different in this study. For example, 33 aroma-active compounds were characterized in the control, and the FD factors of ethanol (alcoholic, 128), 3-methyl-1-butanol (whiskey, malty, burnt, 128), ethyl isovalerate (sweet, fruity, 128), phenylethanol (flower, honey, 128), 3-(methylthio)propanal (cooked potato, 128) and 4-ethylguaiacol (smoky, 128) were the highest, accompanied by 2-methylbutanal (malty, 64), 3-methylbutanal (malty, 64), ethyl acetate (pineapple, 64), 4-vinylguaiacol (smoky, pungent, 64), 2-methylbutanoic acid (sour, smelly, 32), 2-methyl-1-butanol (wine, onion, 32), benzeneacetaldehyde (flower, sweet, 32), HEMF (caramel-like, sweet, 32), etc. However, 43 aroma-active compounds were characterized in the HHP400–30 group, with the highest score in sensory evaluation, among which 3-(methylthio)propanal (512) had the highest FD factor, accompanied by 3-methylbutanal (256), 3-methyl-1-butanol (128), 2-methylbutanal (malty, 128), ethyl isovalerate (128), ethanol (64), phenylethanol (64), benzeneacetaldehyde (64), (E)-β-damascenone (apple, rose, honey, 64), HEMF (64), 4-ethylguaiacol (64), 2-methylbutanoic acid (32), 3-methylbutanoic acid (sour, smelly, 32), ethyl acetate (32), 2-methylpropanal (malty, 32), HDMF (caramel-like, sweet, 32), 4-vinylguaiacol (32), 1-octen-3-ol (mushroom, 32), etc. In addition, compounds such as 3-methylbutanoic acid, ethyl 3-(methylthio) propionate, 2-methyl pyrazine, trimethyl pyrazine and (E)-β-damascenone were detected only in the HHP treatment group, or the FD value was higher in the HHP treatment group, which indicates that HHP treatment can accelerate the formation of such compounds.

In light of the aroma properties described in Table 1, the aroma-active compounds listed in Table 2 were divided into ‘alcoholic’, ‘sour’, ‘malty’, ‘floral’, ‘fruity’, ‘caramel-like’, ‘smoky’ and ‘other aroma’. The total FD factors of aroma-active compounds with ‘sour’, ‘malty’, ‘floral’, ‘caramel-like’ and ‘other aroma’ in the HHP400–30 group were 81, 416, 201, 646 and 55, respectively; these were 80.0%, 225.0%, 24.8%, 251.1% and 358.3% higher than those found in the control group, respectively. However, the total FD factors of compounds with ‘alcoholic’, ‘fruity’ and ‘smoky’ in the HHP400–30 group were 208, 192 and 97, respectively; these were 27.8%, 12.7% and 50.0% lower than those in found the control group, respectively. Based on the above analysis, HHP treatment increased the FD factors of five aromatic compounds in soy sauce, indicating that it may promote aroma formation in raw soy sauce.

### 3.4. OAVs of Aroma-Active Compounds

In order to further elucidate the effect of HHP treatment on the aroma characteristics of soy sauce, OAVs of aroma-active compounds in the control and samples were measured and calculated in this study (Table 2), among which OAVs ≥ 1 were considered to be the main aroma-active compounds due to their relatively high contributions to the overall aroma. Here, 35 main aroma-active compounds were characterized. As exhibited in Table 2, compared with the control, OAVs of ‘sour’, ‘malty’, ‘floral’, ‘caramel-like’ and ‘other aroma’ under the HHP treatment were all improved, except for ‘sour’ and ‘floral’ from the 600 MPa treatment group. In addition, OAVs of ‘malty’ from the 600 MPa treatment group were not higher than those in the 200 MPa and 400 MPa treatment groups. For example, for the malty aroma, the OAV of the control was 551; the OAVs of the 200 MPa treatment groups were 684, 633 and 632, respectively; the 400 MPa treatment groups were 730, 755 and 752, respectively; and those of the 600 MPa group were 576, 573 and 567, respectively. These results indicate that using an appropriate treatment pressure can contribute to the formation of some compounds, while excessive treatment pressure can destroy or affect the formation of other compounds. In addition, the OAVs of ‘alcoholic’, ‘fruity’ and ‘smoky’ were all decreased by the HHP treatment. This is an inevitable consequence of reductions in the contents of related compounds.

The highest OAV score came from the HHP400–30 group on sensory evaluation. Compared with the control, the OAVs of ‘sour’, ‘malty’, ‘floral’, ‘caramel-like’ and ‘other aroma’ in HHP400–30 increased by 19.4%, 37.1%, 36.7%, 49.2% and 117.9%, respectively, while that of ‘alcoholic’, ‘fruity’ and ‘smoky’ decreased by 5.1%, 11.6% and 16.7%, respectively. Overall, the OAV changes observed for main aroma-active compounds were consistent with the sensory evaluation results in Figure 1.

### 3.5. Hierarchical Cluster Analysis

HCA is performed to visually and objectively elucidate the aroma similarity among the samples and control. HCA is performed according to changes in OAVs. As shown in Figure 2, HHP200–10, HHP200–30 and HHP200–60; HHP400–10, HHP400–30 and HHP400–60; and HHP600–10, HHP600–30 and HHP600–60 were firstly grouped into three sets in parallel; the control was a separate group. The earlier the samples were grouped into one cluster, the more similar aromas they had. These results revealed that the effect of treatment pressure on flavor was greater than that of treatment time. Secondly, the control and groups of HHP200–10, HHP200–30 and HHP200–60 were merged into one bigger group (C, HHP200–10, HHP200–30 and HHP200–60), while the 400 MPa and 600 MPa treatment groups remained as two separate groups. This indicated that the aroma of soy sauce treated with 200 MPa was closer to that of the control. Lastly, the group of HHP400–10, HHP400–30 and HHP400–60 and group of C, HHP200–10, HHP200–30 and HHP200–60 were combined into one larger group, while the 600 MPa treatment group remained as a separate group. As demonstrated in Figure 1, the total scores of HHP400–10, HHP400–30 and HHP400–60 were all higher than that of the control, and HHP400–30 received the highest score. The objectivity of the sensory evaluation was substantiated by HCA results. Thus, it could be concluded that appropriate HHP treatment could be used to accelerate aroma formation and significantly improve aroma quality in comparison to the control, but using excessively high pressure could have adverse effects on the aroma of soy sauce.

### 3.6. Principal Component Analysis

PCA was used to further determine the directional alterations of the main aroma-active compounds under different treatment conditions. In this study, 35 OAVs of main aroma-active compounds were used as variables for PCA. The principal component biplot is shown in Figure 3, where two main PCs accounted for 86.7% of the variability. The first principal component (PC1) and the second principal component (PC2) were responsible for 57.8% and 28.9% of the total difference, respectively. The PC1 and PC2 divided soy sauce samples and the control into four different groups. The untreated soy sauce was on the negative side of the PC1 and the PC2; those treated at 200 MPa were on the positive side of the PC2 and negative side of the PC1; those treated at 400 MPa were on the positive side of the PC1 and the PC2; and those treated at 600 MPa were on the positive side of the PC1 and negative side of the PC2. We also found that the processing pressures and processing times of soy sauce were positively correlated with the PC1 except for the HHP600–60 group.

The loadings indicate the relative Importance of each volatile compound to the sample distribution. In general, the distribution of different aroma-active compounds will reflect the difference in soy sauce under different HHP conditions. For example, the positive axis of the PC1 was greatly affected by phenylacetic acid, ethyl 3-methylthiopropionate, 1-octen-3-ol, trimethyl pyrazine, 2,6-dimethyl pyrazine, 2-furanmethanol, acetic acid and dimethyl trisulfide. It indicates that HHP treatment was beneficial to the formation of these compounds. HHP400–30, which has the highest sensory evaluation score, distributed near the positive axis of the PC1 and the PC2, and is highly affected by 2-methylbutanal, 2-methylpropanal, 3-methyl-1-butanol, 3-methylbutanal, methyl benzoate, benzeneacetaldehyde, (E)-β-damascenone, 2-phenylethyl acetate, 3-(methylthio)propanal, HDMF, HEMF, 2-methylbutanoic acid, butanoic acid and 3-methylbutanoic acid. In addition, 3-(methylthio)-1-propanol, 2-methylpropanoic acid, ethyl propanoate, ethyl isovalerate, benzoic acid, ethanol and 4-ethylguaiacol also contribute greatly to its aroma. These compounds have alcoholic aroma, fruity aroma, floral aroma, caramel-like aroma, sour aroma, malty aroma and smoky aroma, respectively, making HHP400–30 the most rich and complex aroma, resulting in the highest sensory evaluation score for the HHP400–30 group.

## 4. Conclusions

In summary, HHP treatment significantly accelerated the formation of soy sauce’s aromatic compounds, and improved its overall aroma by regulating the Maillard, oxidation and hydrolysis reactions. Soy sauce that was treated at 400 MPa for 30 min had stronger attributes of floral, caramel-like, sour and malty aromas, but weaker attributes of alcoholic, fruity and smoky aromas; this particular treatment obtained the highest sensory score. Therefore, HHP technology has great potential to be applied in the production of soy sauce, in order to yield richer aromas and a high-quality product. Further in-depth research on the biochemical mechanisms of HHP responsible for accelerating the formation of soy sauce aroma and improving the aroma of soy sauce is in progress.

## Figures and Tables

**Figure 1 foods-11-02190-f001:**
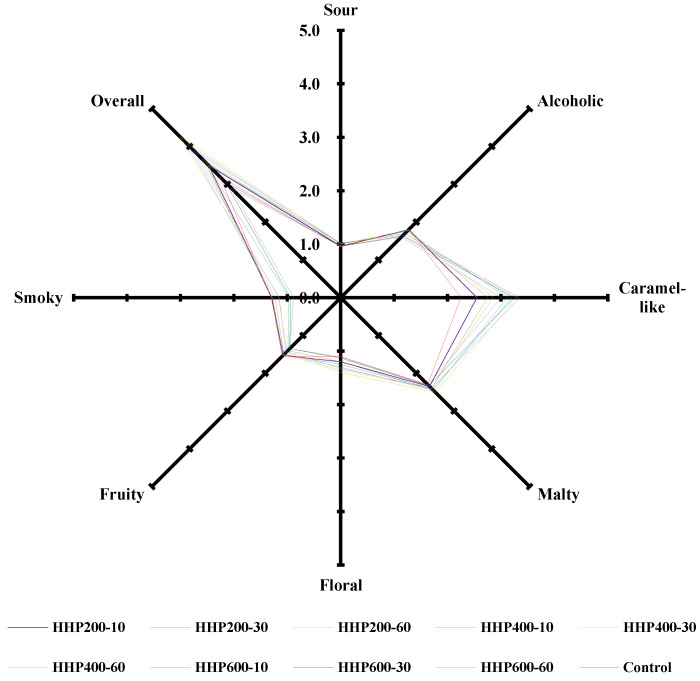
Sensory evaluation of the control and samples.

**Figure 2 foods-11-02190-f002:**
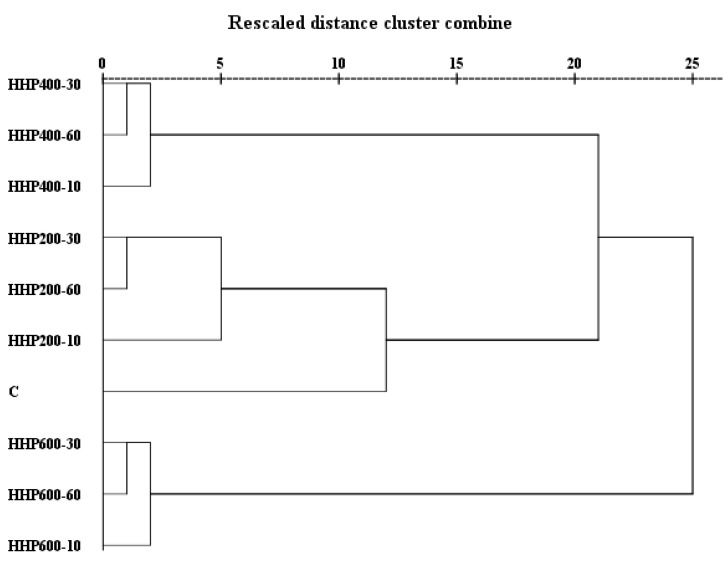
Hierarchical cluster analysis of the control and samples.

**Figure 3 foods-11-02190-f003:**
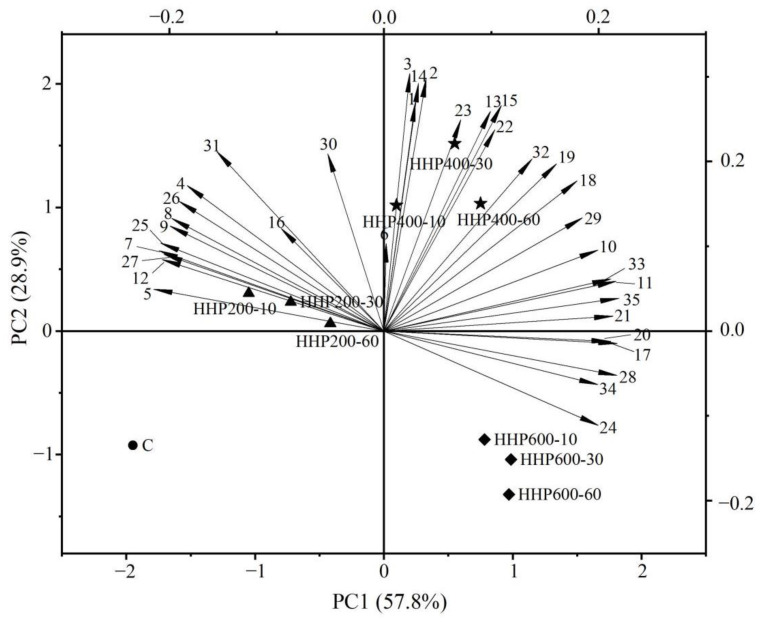
PCA of the control and samples. Annotation: (1) 2-methylpropanal; (2) 2-methylbutanal; (3) 3-methylbutanal; (4) ethanol; (5) 2-methyl-1-butanol; (6) 3-methyl-1-butanol; (7) ethyl acetate; (8) ethyl propanoate; (9) ethyl isovalerate; (10) benzoic acid; (11) ethyl 3-methylthiopropionate; (12) phenylethanol; (13) benzeneacetaldehyde; (14) methyl benzoate; (15) (E)-β-damascenone; (16) 2-phenylethyl acetate; (17) 2-furanmethanol; (18) 3-methylthio-1-propanol; (19) 3-(methylthio)propanal; (20) 2,6-dimethylpyrazine; (21) trimethyl pyrazine; (22) HDMF; (23) HEMF; (24) 2-acetylpyrroline; (25) 4-ethylphenol; (26) 4-ethylguaiacol; (27) 4-vinylguaiacol; (28) acetic acid; (29) 2-methylpropanoic acid; (30) butanoic acid; (31) 2-methylbutanoic acid; (32) 3-methylbutanoic acid; (33) phenylacetic acid; (34) dimethyl trisulfide; (35) 1-octen-3-ol.

**Table 1 foods-11-02190-t001:** Volatile compounds identified in the control and samples.

Compounds	RI (DB-Wax)	Aroma Description	FD										Identification Method
			Control	HHP200-10	HHP200-30	HHP200-60	HHP400-10	HHP400-30	HHP400-60	HHP600-10	HHP600-30	HHP600-60	
** *Acids* **													
Acetic acid	1452	Sour ^1,2,3^	2	2	2	2	4	4	4	8	8	8	ABC
2-Methylpropanoic acid	1543	Sour, rancid ^1,2^	1	2	2	2	4	4	4	4	4	4	ABC
Butanoic acid	1622	Sour, sweaty ^1,2,3^	nd	1	1	–	1	1	–	–	–	–	ABC
2-Methylbutanoic acid	1667	Sour, smelly ^1,2,3^	32	32	32	32	32	32	16	8	8	8	ABC
3-Methylbutanoic acid	1657	Sour, smelly ^1,2,3^	8	16	16	16	32	32	16	32	32	32	ABC
4-Methylpentanoic acid	1803	Sour, smelly 2	nd	–	–	–	–	–	–	–	–	–	A
Hexanoic acid	1825	Sweat, pungent 2	nd	–	–	–	–	–	–	–	–	–	AB
2-Ethylhexanoic acid	1908	un	–	–	–	nd	–	–	–	–	–	–	A
2-Methyl-2-butenoic acid	1869	Pungent 3	nd	–	–	–	–	–	–	–	–	–	A
Nonanoic acid	2202	Green, fat 3	–	nd	–	–	–	–	–	nd	–	–	A
Decanoic acid	2351	Rancid, fat 3	nd	–	–	–	–	–	–	–	–	–	A
Benzoic acid	2410	Flower, fruity 1,2,3	4	4	8	8	16	16	16	8	8	8	AB
Phenylacetic acid	2546	Sour, honey ^1,2,3^	2	4	4	4	8	8	8	8	8	8	ABC
** *Alcohols* **													
Ethanol	928	Alcoholic ^1,2,3^	128	128	128	128	64	64	64	32	32	32	ABC
1-Propanol	1042	Alcoholic, pungent ^3^	–	–	–	–	–	–	–	–	–	–	A
2-Methylpropanol	1095	Wine, solvent, bitter ^3^	–	–	–	–	–	–	–	–	–	–	A
1-Butanol	1146	Medicine, fruit ^3^	–	nd	–	–	–	–	–	–	–	–	AB
2-Butanol	1027	Wine ^3^	nd	–	–	–	–	–	–	–	–	–	AB
2-Methyl-1-butanol	1212	Wine, onion ^1,2,3^	32	16	16	16	16	16	8	8	8	8	ABC
3-Methyl-1-butanol	1206	Whiskey, malty, burnt ^1,2,3^	128	128	128	64	128	128	64	128	64	64	ABC
2,3-Butanediol	1581	Fruit, onion ^2,3^	–	–	–	–	–	–	–	–	–	–	A
2-Pentanol	1121	Green ^3^	–	–	–	–	–	–	–	–	–	–	A
Prenol	1129	Herb ^3^	nd	–	–	–	–	–	–	–	–	–	A
4-Methyl-2-pentanol	1147	un	–	–	–	–	nd	–	–	–	–	–	A
2,3-Dimethylpentanol	1187	un	nd	–	–	–	–	–	–	–	–	–	A
1-Hexanol	1364	Resin, flower, green ^3^	–	–	–	–	–	–	–	–	–	nd	A
4-Methyl-1-hexanol	1263	Sweat ^3^	nd	nd	–	–	–	–	–	–	–	–	A
2-Ethyl-1-hexanol	1482	Rose ^2^	–	–	–	–	–	–	–	–	–	–	A
2-Heptanol	1277	Mushroom ^3^	–	–	–	–	–	–	–	nd	–	–	A
6-Methyl-2-heptanol	1312	un	nd	–	–	–	–	–	–	–	–	–	A
3-Octanol	1343	Nut, mushroom ^3^	–	–	–	–	–	–	–	–	–	–	AB
1-Octen-3-ol	1468	Mushroom ^1,2,3^	8	16	16	32	32	32	32	32	32	32	ABC
1-Nonanol	1506	Fat, green ^3^	–	–	–	–	–	–	–	–	–	–	A
2-Nonanol	1537	Cucumber ^3^	nd	–	–	–	–	–	–	–	–	–	A
Benzyl alcohol	1867	Sweet, flower ^3^	–	–	–	–	–	–	–	–	–	–	AB
Methylbenzyl alcohol	1912	Flower ^3^	nd	–	–	–	–	–	–	–	–	–	A
Phenylethanol	1894	Flower, honey ^1,2,3^	128	64	64	64	64	64	64	32	32	32	ABC
α-Ethyl phenylethanol	2013	un	–	–	–	–	–	–	–	–	–	–	A
1-(2-Butoxyethoxy)-ethanol	1799	un	nd	–	–	–	–	–	–	–	–	–	A
4-Phenyl-3-buten-2-ol	nd	un	nd	–	–	–	–	–	–	–	–	nd	A
** *Aldehydes* **													
Acetaldehyde	718	Pungent, ether ^3^	–	–	–	–	–	–	–	–	–	–	AB
2-Methylpropanal	826	Malty ^1,2,3^	–	32	–	–	32	32	32	–	–	nd	ABC
2-Methylbutanal	934	Malty ^1,2,3^	64	64	64	64	64	128	128	64	64	64	ABC
3-Methylbutanal	925	Malty ^1,2,3^	64	128	128	128	256	256	128	128	128	128	ABC
Benzaldehyde	1500	Almond, caramel-like ^2,3^	–	–	–	–	–	–	–	–	–	–	AB
Benzeneacetaldehyde	1625	Flower, sweet ^1,2,3^	32	32	64	64	64	64	64	64	32	32	ABC
3-Furaldehyde	1453	Bread, sweet, almond ^3^	–	1	2	–	2	4	4	4	4	2	ABC
Methyl cinnamaldehyde	1946	Cinnamon, sweet ^3^	nd	–	–	–	–	–	–	–	–	nd	A
** *Ketones* **													
Acetone	869	Mild ^3^	nd	–	–	–	–	–	–	–	–	–	AB
2-Butanone	943	Cheese ^2,3^	–	–	–	–	1	1	1	1	1	–	A
Acetoin	1288	Butter, creamy ^3^	–	–	–	–	–	–	–	–	–	–	A
Methyl isobutyl ketone	1007	Fruity ^3^	nd	–	–	–	–	–	–	–	–	–	A
2,3-Pentanedione	1058	Butter, creamy ^3^	–	–	–	–	–	–	–	–	–	–	A
3-Penten-2-one	1099	un	nd	–	–	–	–	–	–	–	–	–	A
5-Methyl-2-hexanone	1138	un	–	–	–	–	–	–	–	–	–	–	A
6-Methyl-2-heptanone	nd	un	nd	–	–	–	–	–	–	–	–	–	A
3-Octanone	1248	Herb, butter, resin ^2,3^	1	1	1	1	1	2	2	–	1	nd	AB
Isophorone	1603	Camphor ^2,3^	nd	–	–	–	–	–	–	–	–	–	A
1-Phenyl-1-propanone	1723	Mild ^3^	–	–	–	–	–	–	–	–	–	–	A
2,6,6-Trimethyl-2,4-cycloheptadien-1-one	1691	un	nd	–	–	–	–	–	–	–	–	–	A
3-Hydroxy-3-phenylbutan-2-one	1789	un	–	–	–	–	–	–	–	–	–	–	A
(E)-β-damascenone	1815	Rose, honey, apple ^1,2,3^	–	16	16	16	64	64	32	16	16	16	ABC
** *Esters* **													
Ethyl acetate	905	Pineapple ^1,2,3^	64	32	32	32	32	32	32	16	16	16	ABC
Ethyl propanoate	979	Fruity ^1,2,3^	16	8	8	8	8	8	8	4	4	4	ABC
Ethyl lactate	1355	Fruity ^2,3^	–	–	–	–	–	–	–	–	–	–	A
Ethyl butanoate	1055	Fruity ^2,3^	–	–	–	–	–	–	–	–	–	–	AB
Ethyl isobutyrate	966	Apple ^2,3^	–	–	–	–	–	–	–	–	–	–	A
Ethyl 2-methylbutyrate	1055	Pineapple ^3^	–	–	–	–	–	–	–	–	–	–	A
Ethyl isovalerate	1061	Sweet, fruity ^1,2,3^	128	128	128	128	128	128	128	64	64	64	ABC
Ethyl hexanoate	1233	Wine, fruity ^2,3^	–	–	–	–	–	–	–	–	–	–	AB
Ethyl isohexanoate	1203	Fruity ^3^	–	–	–	nd	–	–	–	–	–	–	A
Ethyl 5-methylhexanoate	1285	un	–	–	–	–	–	nd	–	–	–	–	A
Ethyl 2-ethylhexanoate	1309	un	–	–	–	–	–	–	–	–	–	–	A
Ethyl heptanoate	1455	Fruity ^2,3^	–	–	–	nd	–	nd	nd	nd	nd	nd	A
Ethyl caprylate	1443	Fruity, fat ^2,3^	–	–	–	–	nd	nd	nd	–	nd	nd	AB
Ethyl nonanoate	1468	Grape ^2,3^	–	–	nd	nd	–	–	nd	nd	nd	nd	A
Ethyl palmitate	2265	Wax ^3^	–	–	–	–	–	–	–	–	–	–	A
Ethyl 9-hexadecenoate	2274	un	–	–	–	–	–	–	–	–	–	–	A
Ethyl oleate	2403	Flower ^3^	–	–	–	nd	nd	nd	nd	nd	–	nd	A
(E)-9-Octadecenoic acid ethyl ester	2391	Flower, fruity, fat ^3^	–	–	nd	nd	–	nd	nd	–	nd	nd	A
Ethyl tiglate	1241	Mushroom ^3^	–	–	–	nd	nd	nd	nd	nd	nd	nd	A
Ethyl benzoate	1649	Fruity, flower ^2,3^	8	8	4	4	4	4	4	4	2	2	AB
Ethyl phenylacetate	1771	Fruity, sweet ^2,3^	–	–	–	–	–	–	–	–	–	–	A
2-Phenylethyl acetate	1827	Rose, honey, tobacco ^1,2,3^	1	1	–	–	1	1	1	1	–	–	ABC
Ethyl 3-phenylpropionate	1908	Flower ^3^	–	–	–	nd	–	nd	nd	nd	–	–	A
Butanedioic acid diethyl ester	1632	Wine, fruity ^3^	–	–	nd	nd	–	nd	nd	nd	nd	nd	A
Dimethyl phthalate	2316	Mild ^3^	–	–	–	–	–	–	–	–	–	–	A
Methyl benzoate	1609	Flower, fruity ^1,2,3^	nd	2	2	–	4	8	8	–	–	–	ABC
Methyl phenylacetate	1712	Honey ^3^	nd	–	–	–	–	–	–	–	–	–	A
Isobutyl acetate	1019	Fruity, banana, apple ^2,3^	–	–	–	–	–	–	–	–	–	–	A
Isoamyl acetate	1127	Fruity, banana ^2,3^	–	–	–	–	–	–	–	–	–	–	A
Isoamyl lactate	1499	Sweet ^3^	nd	–	–	–	–	–	–	–	–	–	A
Formic acid heptyl ester	1483	Flower, fruity, fat ^3^	nd	–	–	–	–	–	–	–	–	–	A
Butyrolactone	1521	Caramel-like, sweet ^2,3^	–	–	–	–	–	–	–	–	–	–	AB
** *Furan(one)s* **													
Furan	1456	Sweet, bread ^2,3^	nd	–	–	–	–	–	–	–	–	–	AB
1-(2-Furanyl)-ethanone	1487	Smoky ^2,3^	nd	–	–	–	–	–	–	–	–	–	A
1-(2-Furanyl)-1-propanone	1511	un	–	–	–	–	–	–	–	–	–	–	A
2,2-Dimethyl-5-isopropyl -furan	1569	un	–	–	–	–	–	–	–	–	–	–	A
Benzofuran	nd	Mild ^3^	–	–	–	–	–	–	–	–	–	–	A
3-Phenylfuran	1839	Green bean ^2^	–	–	–	–	1	2	2	2	2	2	A
2-Furanmethanol	1125	Caramel-like ^1,2,3^	1	2	2	2	2	2	2	2	2	2	ABC
Dihydro-5-pentyl-2(3H) -furanone	1971	Coconut, peach ^3^	nd	–	–	–	–	–	–	–	–	–	A
HDMF	2032	Caramel-like, sweet ^1,2,3^	16	32	32	32	32	32	32	32	16	16	ABC
HEMF	2089	Caramel-like, sweet ^1,2,3^	32	32	32	32	64	64	64	32	32	32	ABC
** *Phenols* **													
Phenol	1485	Phenol ^3^	–	–	–	–	–	–	–	–	–	–	AB
4-Ethylphenol	2158	Smoky ^2,3^	2	2	2	1	1	1	1	1	1	1	A
3-Tert-butyl-phenol	2243	Phenol ^3^	–	–	–	–	–	–	–	–	–	–	A
2,4-Di-tert-butylphenol	2289	Phenol ^3^	–	–	–	–	–	–	–	–	–	–	A
4-Ethylguaiacol	2016	Smoky ^1,2,3^	128	128	128	128	64	64	64	64	32	32	ABC
4-Vinylguaiacol	2177	Pungent, smoky ^1,2,3^	64	64	64	32	32	32	32	32	32	32	ABC
5-Methylguaiacol	1932	Sweet ^3^	nd	–	–	–	–	–	–	–	–	–	A
** *Pyrazines* **													
Pyrazine	1205	Rancid ^3^	nd	–	–	–	–	–	–	–	–	–	A
2-Methyl pyrazine	1266	Popcorn ^2,3^	nd	1	1	1	1	2	2	2	2	2	AB
2,3-Dimethyl pyrazine	1259	Roasted nut, cocoa ^3^	nd	–	–	–	–	–	–	–	–	–	A
2,5-Dimethyl pyrazine	1255	Roasted nut, cocoa ^3^	–	–	–	–	–	–	–	–	–	–	A
2,6-Dimethyl pyrazine	1329	Roasted nut, cocoa ^2,3^	–	–	–	8	16	16	16	16	16	16	AB
2-Isopropyl pyrazine	1353	un	nd	–	–	–	–	–	–	–	–	–	A
Trimethyl pyrazine	1388	Roast, potato ^2,3^	1	2	2	2	4	4	4	4	4	4	A
2-Ethyl-6-methyl pyrazine	1345	Fruity, sweet ^3^	–	–	–	–	1	2	2	2	4	2	A
2-Ethyl-3,5-dimethyl pyrazine	1453	Potato ^3^	–	–	–	–	–	–	–	–	–	–	A
** *Sulfur-containing compounds* **													
Disulfide dimethyl	1068	Onion, cabbage, putrid ^2,3^	1	1	2	2	2	4	4	4	4	4	AB
Dimethyl trisulfide	1372	Sulfur, fish, cabbage ^1,2,3^	2	–	4	4	4	8	8	8	8	8	ABC
3-Methylthio-1-propanol	1718	Sweet, potato ^1,2,3^	2	4	4	4	8	8	8	8	4	4	ABC
3-(Methylthio)propanal	1423	Cooked potato ^1,2,3^	128	256	256	256	256	512	512	256	256	256	ABC
2-Hydroxyethyl isobutyl sulfide	1583	un	–	–	–	–	–	–	–	–	–	–	A
Ethyl 3-(methylthio) propionate	1417	Onion, garlic, pineapple ^2^	8	8	4	4	4	4	4	4	2	2	A
3-(Methylthio) propyl acetate	1583	Mushroom, onion, garlic ^2^	–	–	–	–	–	–	–	–	–	–	A
** *Others* **													
2-Acetylpyrroline	1396	Nut, roasted, bread ^1,2,3^	4	4	4	4	8	8	8	8	16	16	AB
3-Methyl-2-pyrrolidinone	nd	un	–	–	–	–	–	–	–	–	–	–	A
Maltol	1952	Caramel-like, sweet ^2,3^	–	–	–	–	–	–	–	–	–	nd	AB
Naphthalene	1725	Mothball-like ^2,3^	–	–	–	–	–	–	–	–	–	–	A
Dodecane	1198	Alkane-like ^3^	–	–	–	–	–	–	–	–	–	–	A
Hexadecane	1599	Alkane-like ^3^	–	–	–	–	–	–	–	–	–	–	A

Aroma descriptions: ^1^ obtained from GC-O in this study; ^2^ taken from references: [3,5,23,24]; ^3^ aroma descriptions based on Flavornet (http://www.flavornet.org/flavornet.html, accessed on 10 February 2022) and TGSC information system (http://www.thegoodscentscompany.com/data/rw1008241.html#toorgano, accessed on 12 February 2022).^un^ unavailable. ^nd^ not detected in GC-MS or GC-O.–the odor was not perceived using GC-O on DB-Wax column. ^A,B,C^ identification methods: ^A^ by matching the mass spectrum of a volatile compound with that in the NIST05 library; ^B^ by comparing the RI and mass spectrum analyzed by GC–MS with those of its standard compound under the same experimental conditions; and ^C^ by comparing the odor quality with that of the corresponding standard compound by GC-O.

**Table 2 foods-11-02190-t002:** OAVs of aroma-active compounds detected in the control and samples.

Aroma-Active Compound	Odor Threshold (μg/L)	Odor Activity Value (OAV)									
		Control	HHP200-10	HHP200-30	HHP200-60	HHP400-10	HHP400-30	HHP400-60	HHP600-10	HHP600-30	HHP600-60
2-Methylpropanal	1.5 ^A^	–	67.1 ± 3.1 ^b^	–	–	82.5 ± 2.9 ^b^	84.1 ± 3.0 ^b^	87.7 ± 3.6 ^a^	–	–	–
2-Methylbutanal	4.4 ^A^	237 ± 13 ^c^	270 ± 10 ^b^	275 ± 12 ^b^	280 ± 12 ^a,b^	287 ± 11 ^a,b^	295 ± 10 ^a^	298 ± 10 ^a^	249 ± 8 ^c^	248 ± 10 ^c^	249 ± 11 ^c^
3-Methylbutanal	1.2 ^A^	314 ± 15 ^d^	347 ± 18 ^b,c^	358 ± 17 ^a,b^	352 ± 17 ^a,b,c^	361 ± 15 ^a,b^	376 ± 13 ^a^	366 ± 16 ^a,b^	327 ± 13 ^c,d^	325 ± 14 ^c,d^	318 ± 11 ^d^
** *OAV of compounds with malty aroma* **		**551**	**684**	**633**	**632**	**731**	**755**	**752**	**576**	**573**	**567**
Ethanol	40,000 ^A^	238 ± 12 ^a,b^	243 ± 10 ^a^	232 ± 11 ^a,b,c^	216 ± 10 ^c,d^	227 ± 10 ^a,b,c,d^	221 ± 10 ^b,c,d^	212 ± 9 ^d^	192 ± 9 ^e^	184 ± 8 ^e^	177 ± 9 ^e^
2-Methyl-1-butanol	16 ^A^	43.3 ± 2.1 ^a^	40.9 ± 1.8 ^a,b^	41.2 ± 1.9 ^a,b^	39.2 ± 1.7 ^b^	35.5 ± 1.5 ^c^	32.6 ± 1.2 ^d^	32.0 ± 1.3 ^d,e^	29.4 ± 1.2 ^e,f^	29.0 ± 1.3 ^f^	29.7 ± 1.2 ^e,f^
3-Methyl-1-butanol	4 ^A^	253 ± 13 ^a^	242 ± 12 ^a^	236 ± 12 ^a^	253 ± 13 ^a^	251 ± 14 ^a^	254 ± 12 ^a^	254 ± 11 ^a^	249 ± 11 ^a^	253 ± 10 ^a^	232 ± 11 ^a^
** *OAV of compounds with alcoholic aroma* **		**534**	**526**	**509**	**508**	**514**	**507**	**498**	**470**	**466**	**439**
Ethyl acetate	5000 ^A^	105 ± 6 ^a^	101 ± 6 ^a,b^	100 ± 5 ^a,b^	97.3 ± 6.1 ^a,b,c^	93.5 ± 5.2 ^b,c^	89.1 ± 4.2 ^c,d,e^	89.6 ± 5.0 ^c,d^	83.7 ± 4.1 ^d,e,f^	80.3 ± 4.2 ^e,f^	78.5 ± 3.9 ^f^
Ethyl propanoate	10 ^A^	16.4 ± 0.8 ^a^	15.8 ± 0.7 ^a^	14.2 ± 0.7 ^b^	13.2 ± 0.7 ^b,c^	12.5 ± 0.6 ^c^	13.6 ± 0.5 ^b^	10.6 ± 0.6 ^d^	8.62 ± 0.43 ^e^	8.51 ± 0.40 ^e^	7.26 ± 0.37 ^f^
Ethyl isovalerate	1.5 ^A^	217 ± 12 ^a^	210 ± 11 ^a^	211 ± 10 ^a^	206 ± 10 ^a^	185 ± 9 ^b^	188 ± 9 ^b^	175 ± 8 ^b,c^	159 ± 8 ^c,d^	147 ± 6 ^d^	145 ± 7 ^d^
Benzoic acid	3000 ^A^	16.6 ± 0.8 ^d^	17.9 ± 0.8 ^c,d^	17.9 ± 0.8 ^c,d^	18.5 ± 0.9 ^c^	19.1 ± 1.0 ^a,b,c^	20.3 ± 1.0 ^a,b^	20.6 ± 1.0 ^a^	19.4 ± 0.9 ^a,b,c^	20.5 ± 0.9 ^a,b^	18.9 ± 0.9 ^b,c^
Ethyl 3-(methylthio) propionate	80 ^C^	–	1.25 ± 0.06 ^d^	1.29 ± 0.05 ^d^	1.47 ± 0.08 ^c^	2.01 ± 0.09 ^b^	2.52 ± 0.12 ^a^	2.55 ± 0.12 ^a^	2.37 ± 0.10 ^a^	2.40 ± 0.11 ^a^	2.50 ± 0.11 ^a^
** *OAV of compounds with fruity aroma* **		**354**	**346**	**344**	**336**	**312**	**313**	**298**	**272**	**259**	**252**
Phenylethanol	564 ^A^	218 ± 11 ^a^	196 ± 9 ^b^	187 ± 9 ^b^	185 ± 8 ^b^	182 ± 7 ^b,c^	187 ± 7 ^b^	170 ± 9 ^c,d^	162 ± 6 ^d,e^	157 ± 6 ^d,e^	155 ± 8 ^e^
Benzeneacetaldehyde	10 ^A^	42.0 ± 2.1 ^d^	46.3 ± 2.3 ^d^	52.8 ± 3.0 ^c^	54.5 ± 3.5 ^c^	69.0 ± 3.2 ^a,b^	71.2 ± 3.5 ^a^	65.1 ± 2.9 ^b^	52.8 ± 2.4 ^c^	53.5 ± 3.0 ^c^	44.6 ± 2.1 ^d^
Methyl benzoate	125 ^A^	–	4.32 ± 0.19 ^c^	4.94 ± 0.23 ^c^	–	7.62 ± 0.35 ^b^	10.2 ± 0.5 ^a^	9.74 ± 0.45 ^a^	–	–	–
(E)-β-damascenone	0.002 ^A^	–	42.8 ± 1.9 ^e^	47.5 ± 2.1 ^d,e^	48.6 ± 2.5 ^d^	59.4 ± 2.9 ^c^	88.2 ± 4.7 ^a^	79.4 ± 4.2 ^b^	32.3 ± 1.3 ^f^	37.2 ± 1.5 ^f^	35.0 ± 1.6 ^f^
2-Phenylethyl acetate	250 ^A^	2.01 ± 0.12 ^a^	1.72 ± 0.08 ^b^	–	–	1.41 ± 0.07 ^c^	1.40 ± 0.06 ^c^	1.35 ± 0.06 ^c^	0.92 ± 0.04 ^d^	–	–
** *OAV of compounds with floral aroma* **		**262**	**291**	**293**	**288**	**319**	**358**	**326**	**248**	**248**	**235**
2-Furanmethanol	2000 ^A^	1.81 ± 0.08 ^e^	2.02 ± 0.09 ^d^	2.11 ± 0.09 ^d^	2.07 ± 0.08 ^d^	2.31 ± 0.11 ^c^	2.57 ± 0.11 ^a,b^	2.54 ± 0.10 ^b^	2.63 ± 0.10 ^a,b^	2.62 ± 0.11 ^a,b^	2.73 ± 0.12 ^a^
3-Methylthio-1-propanol	856.1 ^A^	0.60 ± 0.04 ^g^	2.05 ± 0.10 ^f^	1.93 ± 0.09 ^f^	1.91 ± 0.08 ^f^	3.08 ± 0.13 ^c^	3.65 ± 0.16 ^a^	3.43 ± 0.15 ^b^	2.43 ± 0.14 ^e^	2.72 ± 0.15 ^d^	2.60 ± 0.12 ^d,e^
3-(Methylthio)propanal	0.5 ^A^	378 ± 21 ^f^	438 ± 20 ^e^	471 ± 24 ^d,e^	457 ± 23 ^e^	519 ± 22 ^b,c^	554 ± 25 ^a,b^	562 ± 26 ^a^	448 ± 23 ^e^	509 ± 27 ^c,d^	478 ± 24 ^c,d,e^
2,6-Dimethyl pyrazine	0.4 ^C^	–	–	–	16.4 ± 0.9 ^e^	26.2 ± 1.3 ^d^	32.1 ± 1.5 ^c^	33.5 ± 1.4 ^b,c^	35.1 ± 1.6 ^a,b^	36.1 ± 1.5 ^a,b^	36.5 ± 1.7 ^a^
Trimethyl pyrazine	0.8 ^C^	2.76 ± 0.13 ^f^	4.69 ± 0.21 ^d^	3.86 ± 0.19 ^e^	4.84 ± 0.23 ^d^	5.64 ± 0.27 ^c^	6.66 ± 0.31 ^a,b^	6.21 ± 0.30 ^b^	6.79 ± 0.35 ^a^	6.36 ± 0.32 ^a,b^	6.78 ± 0.33 ^a^
HDMF	25 ^A^	49.8 ± 2.4 ^e^	63.1 ± 3.2 ^b,c^	65.6 ± 3.0 ^a,b^	65.2 ± 3.1 ^a,b,c^	69.0 ± 3.2 ^a^	70.9 ± 3.5 ^a^	69.4 ± 3.2 ^a^	67.1 ± 3.4 ^a,b^	57.3 ± 2.9 ^d^	59.9 ± 3.0 ^c,d^
HEMF	20 ^A^	93.3 ± 4.5 ^d^	90.4 ± 3.7 ^d^	93.1 ± 3.6 ^d^	98.5 ± 4.8 ^c,d^	105 ± 6 ^b,c^	116 ± 5^a^	109 ± 5 ^a,b^	90.6 ± 4.2 ^d^	92.1 ± 4.6 ^d^	95.3 ± 4.9 ^d^
2-Acetylpyrroline	0.1 ^A^	8.29 ± 0.41 ^e^	8.96 ± 0.40 ^e^	9.04 ± 0.44 ^e^	9.16 ± 0.45 ^d,e^	10.0 ± 0.4 ^d^	11.9 ± 0.5 ^c^	11.4 ± 0.5 ^c^	12.9 ± 0.8 ^b^	14.4 ± 0.7 ^a^	14.0 ± 0.7 ^a^
** *OAV of compounds with caramel-like aroma* **		**535**	**609**	**646**	**655**	**741**	**798**	**798**	**666**	**720**	**696**
4-Ethylphenol	140 ^B^	2.79 ± 0.15 ^a^	2.58 ± 0.13 ^b^	2.43 ± 0.13 ^b,c^	2.31 ± 0.12 ^c^	2.04 ± 0.10 ^d^	1.88 ± 0.09 ^d^	1.53 ± 0.06 ^e^	1.25 ± 0.04 ^f^	1.11 ± 0.05 ^f^	0.86 ± 0.04 ^g^
4-Ethylguaiacol	10 ^A^	307 ± 18 ^a^	287 ± 16 ^a,b^	262 ± 13 ^c,d^	268 ± 12 ^b,c^	269 ± 13 ^b,c^	259 ± 12 ^c,d^	242 ± 12 ^d^	202 ± 11 ^e^	182 ± 9 ^e,f^	179 ± 9 ^f^
4-Vinylguaiacol	12 ^A^	90.1 ± 4.7 ^a^	78.8 ± 4.1 ^b^	76.1 ± 3.9 ^b^	68.3 ± 3.5 ^c,d^	66.9 ± 2.7 ^d,e^	72.9 ± 3.2 ^b,c^	61.3 ± 3.0 ^e,f^	56.8 ± 2.3 ^f,g^	53.9 ± 2.8 ^g,h^	50.3 ± 2.1 ^h^
** *OAV of compounds with smoky aroma* **		**400**	**368**	**340**	**339**	**338**	**334**	**305**	**260**	**238**	**231**
Acetic acid	10000 ^A^	10.5 ± 0.6 ^e^	14.6 ± 0.9 ^d^	14.7 ± 0.8 ^d^	15.5 ± 0.8 ^d^	18.5 ± 1.0 ^c^	20.1 ± 1.2 ^b,c^	21.8 ± 1.0 ^b^	23.7 ± 1.5 ^a^	24.0 ± 1.3 ^a^	24.5 ± 1.3 ^a^
2-Methylpropanoic acid	3500 ^A^	9.61 ± 0.45 ^d^	11.8 ± 0.7 ^c^	12.2 ± 0.4 ^b,c^	12.9 ± 0.6 ^a,b,c^	13.1 ± 0.6 ^a,b^	13.9 ± 0.5 ^a^	13.7 ± 0.8 ^a^	13.1 ± 0.5 ^a,b^	13.8 ± 0.5 ^a^	12.2 ± 0.6 ^b,c^
Butanoic acid	173 ^A^	–	7.25 ± 0.32 ^a^	7.53 ± 0.27 ^a^	–	7.31 ± 0.51 ^a^	7.64 ± 0.35 ^a^	–	–	–	–
2-Methylbutanoic acid	225 ^A^	150 ± 8 ^a^	146 ± 8 ^a,b^	149 ± 7 ^a,b^	147 ± 8 ^a,b^	141 ± 6 ^a,b^	137 ± 7 ^b^	138 ± 7 ^a,b^	57.4 ± 3.4 ^c^	52.5 ± 3.1 ^c^	60.4 ± 3.7 ^c^
3-Methylbutanoic acid	540 ^A^	15.8 ± 0.9 ^d^	34.6 ± 2.0 ^c^	39.0 ± 1.5 ^b^	39.7 ± 1.7 ^b^	43.8 ± 2.4 ^a^	44.0 ± 2.2 ^a^	45.2 ± 1.9 ^a^	40.0 ± 2.1 ^b^	32.2 ± 1.7 ^c^	35.4 ± 1.8 ^c^
Phenylacetic acid	1000 ^A^	10.2 ± 0.5 ^b^	11.2 ± 0.5 ^a^	11.6 ± 0.7 ^a^	11.6 ± 0.6 ^a^	11.9 ± 0.5 ^a^	12.1 ± 0.4 ^a^	12.2 ± 0.7 ^a^	12.1 ± 0.5 ^a^	12.1 ± 0.6 ^a^	12.2 ± 0.4 ^a^
** *OAV of compounds with sour aroma* **		**196**	**225**	**234**	**227**	**235**	**235**	**231**	**146**	**135**	**145**
Dimethyl trisulfide	0.01 ^A^	4.13 ± 0.25 ^f^	–	7.16 ± 0.40 ^e^	6.39 ± 0.35 ^e^	8.77 ± 0.49 ^d^	10.8 ± 0.5 ^c^	12.1 ± 0.7 ^b^	13.9 ± 0.5 ^a^	13.3 ± 0.6 ^a^	14.0 ± 0.8 ^a^
1-Octen-3-ol	1.5 ^A^	21.2 ± 1.7 ^d^	30.4 ± 1.4 ^c^	32.9 ± 2.5 ^c^	37.7 ± 1.5 ^b^	40.2 ± 1.7 ^b^	44.4 ± 2.3 ^a^	45.2 ± 2.0 ^a^	45.0 ± 2.0 ^a^	45.2 ± 2.1 ^a^	44.6 ± 2.3 ^a^
** *OAV of compounds* ** ** *with other aroma* **		**25.3**	**30.4**	**40.1**	**44.1**	**48.9**	**55.1**	**57.3**	**58.9**	**58.5**	**58.6**

Odor thresholds were taken from references: ^A^ [3], ^B^ [35] and ^C^ determination by our laboratory. ^a–j^ Different letters in the same row indicate significant differences (*p* < 0.05).–not detected in GC-MS.

## Data Availability

Data is contained within the article or Appendix A.

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
