# Peer review of "Improving Soy Sauce Aroma Using High Hydrostatic Pressure and the Preliminary Mechanism"

_foods, 2022, doi:10.3390/foods11152190_

Round 1

Reviewer 1 Report

I have reviewed the manuscript entitled " Improving Soy Sauce Aroma Using High Hydrostatic Pressure and the Preliminary Mechanism”. This paper assesses the influences of HHP treatment on the aroma compounds and aroma of soy sauce and elucidates the preliminary mechanism of the rapid aroma maturation of soy sauce caused by HHP treatment. The paper is well presented and easy to read. The introduction provides a good, generalized background of the topic that quickly gives the reader an appreciation of the wide range of applications for soy sauce key odorants. The literature cited is relevant to the study. I think the paper could prove to be very interesting and useful to very large researchers, after some revisions.

My remarks about the text are as follows:

Line 29: Aspergillus oryze was changed as Aspergillus oryzae

Line 81: Add the purchasing year of the samples.

Line 89: 4-ethylguaiacoll was changed as 4-ethylguaiacol

Line 122: SPME fiber length 1 or 2 cm.

Line 125: Add the stirring rpm.

Line 133: Identification took place by using only one DB-Wax capillary column. For proper verification of identification, it is necessary to compare retention indices on two columns of different polarity. In most cases substances, which co-elute on one column can be distinguished on the other column with different polarities. Please explain

Line 146: Add a reference here: The odor activity value (OAV) of each aroma-active compound was calculated using the corresponding threshold value to divide its concentration (Guclu et al., 2016).

Guclu, G.; Sevindik, O.; Kelebek, H.; Selli, S. Determination of Volatiles by Odor Activity Value and Phenolics of cv. Ayvalik Early-Harvest Olive Oil. Foods 2016, 5, 46. https://doi.org/10.3390/foods5030046

Pang, X., Guo, X., Qin, Z., Yao, Y., Hu, X., & Wu, J. (2012). Identification of aroma-active compounds in Jiashi muskmelon juice by GC-O-MS and OAV calculation. Journal of Agricultural and Food Chemistry, 60(17), 4179-4185.

In figure 1: Over-all was changed as Overall

Line 209: 3-(methothio) propanol was changed as 3-(methylthio) propanol

Author Response

Response to Reviewers Comments

We would like to thank the editor and reviewers for your hard work and insightful comments, which have helped us to significantly improve the quality of our manuscript. We have revised the manuscript according to your suggestions as described in our point by point responses below (The revised parts were marked in red in the manuscript).

Reviewer 1:

Comments and Suggestions for Authors

I have reviewed the manuscript entitled "Improving Soy Sauce Aroma Using High Hydrostatic Pressure and the Preliminary Mechanism”. This paper assesses the influences of HHP treatment on the aroma compounds and aroma of soy sauce and elucidates the preliminary mechanism of the rapid aroma maturation of soy sauce caused by HHP treatment. The paper is well presented and easy to read. The introduction provides a good, generalized background of the topic that quickly gives the reader an appreciation of the wide range of applications for soy sauce key odorants. The literature cited is relevant to the study. I think the paper could prove to be very interesting and useful to very large researchers, after some revisions.

My remarks about the text are as follows:

Response: Thanks for the reviewer’s positive comments on our work.

(1) Line 29: Aspergillus oryze was changed as Aspergillus oryzae.

Response: Thanks a lot for your reminder. We have replaced “Aspergillus oryze” with “Aspergillus oryzae” in line 32.

(2) Line 81: Add the purchasing year of the samples.

Response: Thanks a lot for your reminder. We have added the purchasing year of the samples in line 88-100.

(3) Line 89: 4-ethylguaiacoll was changed as 4-ethylguaiacol.

Response: Thanks a lot for your reminder. We have replaced “4-ethylguaiacoll” with “4-ethylguaiacol” in line 95.

(4) Line 122: SPME fiber length 1 or 2 cm.

Response: Thanks a lot for your reminder. The SPME Fiber model we selected is 75 μm CAR/PDMS SPME Fiber, 75 μm here refers to the thickness of carboxen-polydimethylsiloxane coating, which has been modified in line 130-131 for easy understanding.

(5) Line 125: Add the stirring rpm.

Response: Thanks a lot for your reminder. We have added the stirring rpm (200 rpm) in line 134.

(6) Line 133: Identification took place by using only one DB-Wax capillary column. For proper verification of identification, it is necessary to compare retention indices on two columns of different polarity. In most cases substances, which co-elute on one column can be distinguished on the other column with different polarities. Please explain.

Response: Thanks a lot for your reminder. A number of previous experiments have shown that most of the aroma compounds can be distinguished on the DB-Wax capillary column (Gao et al., 2010; Gao et al., 2018; Gao et al., 2010; Gao et al., 2020; Gao et al., 2022) , so we only selected one DB-Wax capillary column in the experiment.

  1. Gao, X. L.; Cui, C.; Zhao, H. F.; Zhao, M. M.; Yang, L.; Ren, J. Y. Changes in volatile aroma compounds of traditional Chinese-type soy sauce during moromi fermentation and heat treatment. Food Sci. Biotechnol. 2010, 19, 889–898.
  2. Gao, X. L.; Zhang, J. K.; Regenstein, J. M.; Yin, Y. Y.; Zhou, C. S. Characterization of taste and aroma compounds in Tianyou, a traditional fermented wheat flour condiment. Food Res. Int. 2018, 106, 156–163.
  3. Gao, X. L.; Liu, E. M.; Zhang, J. K.; Yang, L. X.; Huang, Q. R.; Chen, S.; Ma, H. L.; Ho, C. T.; Liao, L. Accelerating aroma maturation of raw soy sauce using low intensity sonication. Food Chem. 2020, 329, 127118.
  4. Gao, X. L.; Shan, P.; Feng, T.; Zhang, L. J.; He, P.; Ran, J. L.; Fu, J. Y.; Zhou, C. S. Enhancing selenium and key flavor compounds contents in soy sauce using selenium-enriched soybean. Food Compos. Anal. 2022, 106, 104299.

(7) Line 146: Add a reference here: The odor activity value (OAV) of each aroma-active compound was calculated using the corresponding threshold value to divide its concentration (Guclu et al., 2016).

  1. Guclu, G.; Sevindik, O.; Kelebek, H.; Selli, S. Determination of Volatiles by Odor Activity Value and Phenolics of cv. Ayvalik Early-Harvest Olive Oil. Foods. 2016, 5, 46. https://doi.org/10.3390/foods5030046
  2. Pang, X., Guo, X., Qin, Z., Yao, Y., Hu, X., & Wu, J. (2012). Identification of aroma-active compounds in Jiashi muskmelon juice by GC-O-MS and OAV calculation. Journal of Agricultural and Food Chemistry, 60(17), 4179-4185.

Response: Thank you very much for your suggestion. We studied the reference carefully and cited it in line 162.

(8) In figure 1: Over-all was changed as Overall.

Response: Thanks a lot for your reminder. We have replaced “Over-all” with “Overall” in figure 1.

(9) Line 209: 3-(methothio) propanol was changed as 3-(methylthio) propanol.

Response: Thanks a lot for your reminder. We have replaced “3-(methothio) propanol” with “3-(methylthio) propanol” in line 228 and 231.

Reviewer 2 Report

The manuscript “Improving Soy Sauce Aroma Using High Hydrostatic Pressure and the Preliminary Mechanism” is generally very well written and contains data of some relevance for a general readers as well as of high relevance for specialists in the topic. Although the subject of the paper could be of interest for the readers of the journal, the paper needs some corrections.

In general, in my opinion the manuscript is very interesting. The analyses performed in the work are very comprehensive. The determination of aromatic compounds is carried out by various methods. The statistical analysis is also well described.

The work requires some minor corrections:

-  - Line 171: Instead of the word "had" it is better to use the word "was characterized by".

-  - Figure 1: I believe it is better to explain what the abbreviation “C” stands for. It would be good to highlight the control sample with a different colour / thickness.

-  - Line 279: A different font was used.

Author Response

Reviewer 2:

Comments and Suggestions for Authors

Dear Authors,

The study shows new application for high hydrostatic pressure. The combination of various evaluation methods contribute to the quality of the study.

Nevertheless, I have some questions for discussion.

Response: Thanks a lot for the reviewer’s positive comments.

(1) In the first line of introduction you indicated umami taste as one of the most important characteristics. I wonder if this is not related to some aroma compounds? In the same paragraph (Line 29) you mentioned yeast and lactic acid bacteria starters, but it was not mentioned any further, so those are not always used?

Response: Thanks a lot for the questions. The umami taste of soy sauce mainly comes from amino acids and small peptides produced by protein decomposition, especially glutamic acid and aspartic acid, and other taste-associated substances, such as nucleotides and salt (Zhuang et al., 2016). At present, there is no evidence to prove the direct relationship between umami taste and aroma compounds, but some studies have shown that valine, isoleucine and leucine are the precursors of aroma compounds (Feng et al., 2013).

Yeast and lactic acid bacteria are artificially added to Japanese-type soy sauce, but not to Chinese-type soy sauce. However, yeast and lactic acid bacteria in the air will fall into koji and participate in the subsequent fermentation process of soy sauce, producing ethanol and lactic acid, respectively.

  1. Zhuang, M. Z.; Lin, L. Z.; Zhao, M. M.; Dong, Y.; Sun-Waterhouse, D.; Chen, H. P.; Qiu, C. Y.; Su, G. W. Sequence, taste and umami-enhancing effect of the peptides separated from soy sauce. Food Chem. 2016, 206, 174–181.
  2. Feng, Y. Z.; Cui, C.; Zhao, H. F.; Gao, X. L.; Zhao, M. M; Sun, W. Z. Effect of koji fermentation on generation of volatile compounds in soy sauce production. Int. J. Food Sci. Tech. 2013, 48, 609–619.

(2) Lines70-71 refers to inactivation of unwanted microorganisms, but what about normal microflora. Would it be possible, that other microflora is also inactivated and less salt could be used in the product?

Response. Thanks a lot for the questions. Our HHP treatment is carried out after the fermentation of soy sauce. At present, HHP treatment cannot be applied during soy sauce fermentation. Both references and laboratory studies have found that normal microfloras in soy sauce can be inactivated above 200 MPa (Buzrul et al., 2004), which is of great help to the subsequent sterilization of soy sauce. But at present, the content of salt in soy sauce fermentation can not be reduced, because salt can inhibit the growth of unwanted microfloras. Therefore, whether HHP can be used to produce low salt soy sauce needs further study.

  1. Buzrul, S.; Alpas, H.; Bozoglu, F. Effects of high hydrostatic pressure on shelf life of lager beer. Eur. Food Res. Technol. 2004, 220, 615–618. 

(3) I doubt if the fermented soy sauce (koji) is the proper control. Now you compare HHP treated sauce with the sauce which is not ready yet. Or possibly I got confused with the theory described.

Response: Thanks a lot for the question. The soy sauce fermented for 30 days was filtered, with different HHP treatments as samples and untreated ones as control. In order to verify the influence of HHP on the aroma of soy sauce, we compared with the previous research results of our team in lines 185-187. The results showed that the sensory evaluation score of soy sauce under the optimal HHP treatment was higher than that of soy sauce after 180 days of fermentation, suggesting that HHP treatment significantly improved the raw soy sauce’s aroma.

(4) Lines 103-106 - was sample simply packed and no vacuum? Was air in the bags? What was temperature increase due to pressurisation? What was rate of pressure increase, what was rate of depressurisation?

Response: Thanks a lot for the questions. The samples were packed in polyethylene bags without vacuuming. After manual air discharge, the bags were sealed, leaving a little air in the bags. Moreover, the pressure increase time was around 40 s for 200 MPa, 60 s for 400 MPa and 80 s for 600 MPa. The temperature in the pressurization process did not exceed 35 ℃. The depressurisation time was 10 s. The key paramters have been added in the manuscript.

(5) Lines 128-131: there is nothing about FD, which is mentioned in the caption.

Response: Thanks a lot for your reminder. We have added this section at line 141-144.

(6) It is difficult to read Figure 1. Could it be possible to make it more clear?

Response: Thanks a lot for your reminder. We have modified Figure 1 as the following:

(7) I am totally confused about Table 1. What represents dash (-)? There are many compounds, which has only "-" or "nd" and the aroma itself is described from literature. Do you really need those compounds include in the table? Additionally what presents the values in the table. For example Acetic acid in Control sample is 2. What it is? Table contains Capital letters for description of methods, but in the note under the table, there are only capital letters in superscript. 

Response: Thanks a lot for the questions. The dash (-) represents odors cannot be noticed by GC-O on DB-WAX column. Although some compounds are denoted by "-", which can be detected by GC-MS, and would have important reference value to the researchers. The compound denoted by "nd" means that they were not detected by either GC-O or GC-MS. We have annotated them below Table 1.

For example, acetic acid in control sample is 2, which means that the highest split ratio of GC injection at which it can be noticed is 2:1, and so on. For the identification methods, “A” means by matching the mass spectrum of a volatile compound with that in the NIST05 library; “B” means by comparing the RI and mass spectrum analyzed by GC-MS with those of its standard compound under the same experimental condition; and “C” means by comparing the odor quality with that of the corresponding standard compound by GC-O.

(8) Lines 198-200 and other places in text Authors refer to numbers of volatiles identified, but when I go back to the table there in many lines are nothing. So, how you can state, that these were detected. Additionally there are places in text which claim (data not shown). I would suggest to add supplementary material with the data, and the you can refer to that.

Line 263 and some other places Authors claim that 7 esters were improved. I assume you can say that their amount increased, but you can not from this study for sure say, that this is improvement. It is just increase in amount, but what it does for sauce sensory properties is not clear.

Response: Thanks a lot for your reminder. We have added supplementary material with the data. As for the increased esters, no adverse effects were found in the sensory evaluation. The specific effect on the aroma of soy sauce needs further study. However, the aroma of soy sauce after HHP treatment was richer and more mellow. Of course, this was not only the effect of esters, but the synergic effect of all the aroma-active compounds.

(9) Line 304: Please check if this is correct, that you have analysed GC-O. I think you have analysed volatiles using GC-O...

Response: Thanks a lot for the question. We have analysed GC-O. Pictures of instruments and standards are shown as following:

(10) Figure 3 is very complex. Could it be possible to make different numbers for compounds, starting from 1 on left side and finishing with 35 on the right. Or reconsider the the abbreviations under the figure, you could give (1) 2-methylpropanal; (2) 2-methylbutanal; (3) 3-methyl....; (4)

Response: Thanks a lot for your suggestion. We have revised it according to the reviewer’s suggestion.

Reviewer 3 Report

Dear Authors,

The study shows new application for high hydrostatic pressure. The combination of various evaluation methods contribute to the quality of the study. 

Nevertheless, I have some questions for discussion.

1. In the first line of introduction you indicated umami taste as one of the most important characteristics. I wonder if this is not related to some aroma compounds? In the same paragraph (Line 29) you mentioned yeast and lactic acid bacteria starters, but it was not mentioned any further, so those are not always used?

2. Lines70-71 refers to inactivation of unwanted microorganisms, but what about normal microflora. Would it be possible, that other microflora is also inactivated and less salt could be used in the product?

3. I doubt if the fermented soy sauce (koji) is the proper control. Now you compare HHP treated sauce with the sauce which is not ready yet. Or possibly I got confused with the theory described.

4. Lines 103-106 - was sample simply packed and no vacuum? Was air in the bags? What was temperature increase due to pressurisation? What was rate of pressure increase, what was rate of depressurisation?

54. Lines 128-131: there is nothing about FD, which is mentioned in the caption.

6. It is difficult to read Figure 1. Could it be posible to make it more clear?

7. I am totally confused about Table 1. What represents dash (-)? There are many compounds, which has only "-" or "nd" and the aroma itself is described from literature. Do you really need those compounds include in the table? Additionally what presents the values in the table. For example Acetic acid in Control sample is 2. What it is? Table contains Capital letters for description of methods, but in the note under the table, there are only capital letters in superscript.

8. Lines 198-200 and other places in text Authors refer to numbers of volatiles identified, but when I go back to the table there in many lines are nothing. So, how you can state, that these were detected. Additionally there are places in text which claim (data not shown). I would suggest to add supplementary material with the data, and the you can refer to that.

Line 263 and some other places Authors claim that 7 esters were improved. I assume you can say that their amount increased, but you can not from this study for sure say, that this is improvement. It is just increase in amount, but what it does for sauce sensory properties is not clear.

9. Line 304: Please check if this is correct, that you have analysed GC-O. I think you have analysed volatiles using GC-O...

10. Figure 3 is very complex. Could it be possible to make different numbers for compounds, starting from 1 on left side and finishing with 35 on the right. Or reconsider the the abbreviations under the figure, you could give (1) 2-methylpropanal; (2) 2-methylbutanal; (3) 3-methyl....; (4) 

Author Response

Reviewer 3:

Comments and Suggestions for Authors

The manuscript“Improving Soy Sauce Aroma Using High Hydrostatic Pressure and the Preliminary Mechanism” is generally very well written and contains data of some relevance for a general readers as well as of high relevance for specialists in the topic. Although the subject of the paper could be of interest for the readers of the journal, the paper needs some corrections.

In general, in my opinion the manuscript is very interesting. The analyses performed in the work are very comprehensive. The determination of aromatic compounds is carried out by various methods. The statistical analysis is also well described.

Response: Thanks a lot for the reviewer’s positive comments.

The work requires some minor corrections:

  • Line 171: Instead of the word "had" it is better to use the word "was characterized by".

Response: Thanks a lot for your reminder. We have replaced “had” with “was characterized by” in line 191.

  • Figure 1: I believe it is better to explain what the abbreviation “C” stands for. It would be good to highlight the control sample with a different colour / thickness.

Response: Thanks a lot for your suggestions. We have replaced “C” with “Control” in Figure 1. We've shown the control sample in red and put the control sample on the top layer.

  • Line 279: A different font was used.

Response: Thanks a lot for your reminder. We have corrected the mistake in line 301.
